# Determining Ion Toxicity in Cucumber under Salinity Stress

**Tsu-Wei Chen *** , **Ilka Mabell Gomez Pineda, Annika Marlen Brand and Hartmut Stützel**

Institute of Horticultural Production Systems, Leibniz Universität Hannover, Herrenhäuser Straße 2, 30419 Hannover, Germany; ilkagpineda@gmail.com (I.M.G.P.); annika.brand@gmx.net (A.M.B.); stuetzel@gem.uni-hannover.de (H.S.)
* Correspondence: chen@gem.uni-hannover.de

**Abstract:** Cucumber (*Cucumis sativus* L.), an important vegetable crop, is sensitive to NaCl. Its salinity tolerance can be improved by grafting onto pumpkin rootstocks, which restricts the uptake of $Na^+$, but not of $Cl^-$. Although $Na^+$ seems to be more toxic than $Cl^-$ in cucumber, tissue tolerance to $Na^+$ and $Cl^-$ is still unclear. In this study, a mixed-salt experiment, designed for equal osmolarity and equimolar concentrations of ions between treatments, was conducted using cucumber genotypes "Aramon" and "Line-759," which are different in $Na^+$ and $Cl^-$ exclusion. This combination of treatments generated various patterns of ion concentrations in leaves for deriving the response curves of photosynthesis and stomatal conductance to ion concentrations. In both cultivars, photosynthesis and stomatal conductance were sensitive to leaf $Na^+$ concentration but insensitive to $Cl^-$ concentration. In these genotypes, tissue tolerance to $Na^+$ varied independently of $Na^+$ exclusion. Grafting "Aramon" onto pumpkin rootstock modified the $Na^+/Cl^-$ ratio in leaves, reduced $Na^+$ uptake, enhanced $K^+$ transport towards the young leaves, and induced $Cl^-$ recirculation to the old leaves. These results suggest that (1) cucumber cannot restrict the $Na^+$ accumulation in leaves but is able to avoid overaccumulation of $Cl^-$, and (2) pumpkin rootstock regulates the recirculation of $K^+$ and $Cl^-$, but not $Na^+$.

**Keywords:** salt stress; grafting; tissue tolerance; ion toxicity; ion recirculation; ion exclusion; *Cucumis sativus*

## 1. Introduction

Salinity stress affects growth and yield of crops worldwide [1]. Excessive salt concentrations in the root zone of crops may result from irrigation with saline water under poor drainage, or the accumulation of salts in closed hydroponic systems [2]. Plant growth under salinity depends on the ability to maintain water uptake from the growth medium [3], to exclude toxic ions entering the shoot [4–6], to redistribute excess ions to senescent leaves [7,8], or any combination of these mechanisms [9].

Cucumber (*Cucumis sativus* L.) is one of the most important and widely grown vegetable crops in the world and is considered salt sensitive. It can tolerate an electrical conductivity of about 2.5 dS m$^{-1}$ and the fruit yield decreases by 13% with each unit increase of electrical conductivity above the threshold value [10,11]. To improve its performance under salinity, genotypic variation in physiological responses to salinity was studied (in cucumber, see [12]). Beside the search for genetic material for breeding salt-tolerant cultivars, grafting cucumber onto pumpkin was demonstrated to be the most successful method to improve salinity tolerance [13–16]. It was reported that the restriction of $Na^+$ uptake by the roots of pumpkin is the most evident physiological mechanism of acquired salinity tolerance [16,17]. Since $Na^+$ may disturb the functions of photosynthetic enzymes [18] and interfere with physiological functions controlled by $K^+$, e.g., stomatal regulation [1,18], the ability of pumpkin

roots to exclude $Na^+$ improves $K^+$ homeostasis and salinity tolerance. However, pumpkin roots do not exclude $Cl^-$ so that grafting does not change the $Cl^-$ contents in cucumber under NaCl salinity [14,15]. Therefore, it was concluded from mixed salt experiments in combination with different grafting types that $Cl^-$ toxicity significantly contributes to yield reduction under salinity [14,15]. However, it was shown that, in cucumber leaves, photosynthesis is better correlated with $Na^+$ concentration than with $Cl^-$ [19]. There are still no convincing data showing the mechanism of $Cl^-$ specific toxicity and the tissue tolerance of $Cl^-$ in cucumber since $Na^+$ and $Cl^-$ concentrations are very well correlated under NaCl salinity [2,5,11,18].

Knowledge about the specific effects of $Na^+$ and $Cl^-$ on physiological traits could only be obtained by using mixed-salt experiments, where the physiological parameters can be compared between NaCl, $Na^+$ dominant, and $Cl^-$ dominant treatments [20–22]. However, treatments in a mixed-salt experiment are only comparable if the osmolarity and ion concentrations are equal for all treatments. It was shown that differences in osmolarity affect ion fluxes into the shoots. For example, salt accumulation rate can be faster in plants grown under low rather than high salinity because a higher transpiration rate under low salinity increases the total ion flux into the leaves due to the smaller osmotic effects on stomatal conductance [19]. As a consequence, photosynthesis could be more restricted under low rather than high salinity due to the stronger ion toxicity [19]. Furthermore, salt concentration in the nutrient solution also affects the salt accumulation rate in leaves due to the nonlinear relationships between ion concentrations in the nutrient solution and in the xylem sap [23–25]. However, in many mixed-salt experiments either equal osmolarity [14,15] or equimolar ion concentrations [20,26,27] were used. Therefore, conclusions drawn from these experiments could be biased due to the differences in ion accumulation rates between treatments. Furthermore, quantifying ion toxicity and tissue tolerance requires response curves of physiological parameters to ion concentrations in leaves [18,28]. This means that both physiological parameters and chemical concentrations must be measured at the same time and repeated many times during the experiment. However, to our knowledge, response curves of this kind have not yet been established in any mixed-salt experiment.

Ion recirculation from young to old leaves is considered an important mechanism for protecting young leaves from ion toxicity [1,7,8]. Under NaCl salinity, young cucumber leaves accumulate less $Cl^-$ but similar $Na^+$, when compared to old leaves [29]. This indicates that cucumber may be able to recirculate $Cl^-$ but not $Na^+$, from young to old leaves. Interestingly, results from an experiment using cucumber grafted onto pumpkin show that $Na^+$ concentration in the xylem sap collected from the rootstocks is higher than from the scions [16]. Therefore, it can be hypothesized that grafting onto pumpkin enhances $Na^+$ recirculation in the shoots as a protective mechanism against ion toxicity. However, this hypothesis remains to be tested.

Here we hypothesize that $Na^+$ is more toxic than $Cl^-$ in cucumber. Two approaches to modify the ratio of $Na^+/Cl^-$ concentration in leaves were used to test this hypothesis. First, a mixed-salt experiment with equal osmolarities and ion concentrations in all treatments was conducted using two different cucumber genotypes, Aramon F1 and Line-759 F1, with different abilities of excluding $Na^+$ and $Cl^-$ [22]. This combination of treatments created various patterns of ion concentrations and ratios in leaves, allowing tests of models to predict the toxic effects of different ions on leaf photosynthesis and stomatal conductance. Second, the ratio of $Na^+/Cl^-$ concentration in leaves was changed by using the genotype Aramon grafted onto pumpkin rootstock. Furthermore, we used the grafting experiment to test the hypothesis that grafting onto pumpkin rootstocks facilitates $Na^+$ recirculation in the cucumber scion.

## 2. Materials and Methods

### 2.1. Experiment 1—Effects of Salt Combinations

Two cucumber (*Cucumis sativus* L.) genotypes, "Aramon" (Rijk Zwaan, De Lier, the Netherlands) and "Line-759" (accession PI 432858, originated from China), were selected for this experiment.

These genotypes were selected because of their differences in ion selectivity [22]. Seeds were sown in rockwool cubes (36 × 36 × 40 mm) in the greenhouse of the Institute of Horticultural Production Systems, Leibniz Universität Hannover, Germany (52.5°N, 9.7°E) on 30 March 2014. Seven days after sowing, seedlings were transplanted into larger rockwool cubes (10 × 10 × 6.5 cm) for another seven days. After that, each seedling was transplanted upon Styrofoam floating in a container with 25 L nutrient solution. Each liter of the control nutrient solution contained 0.53 g $Ca(NO_3)_2$ and 0.65 g Ferty Basisdünger 1 (Planta GmbH, Regenstauf, Germany, 14% $P_2O_5$, 38% $K_2O$, 5% MgO; the solution contained 5.3 mM $K^+$, 1.5 mM $Na^+$, 3.0 mM $Ca^{2+}$, 0.8 mM $Mg^{2+}$, 1.3 mM $H_2PO_4$ 5.9 mM $NO_3^-$, 1.5 mM $Cl^-$, as well as adequate amounts of the micronutrients).

On 26 April 2014, when the third leaves were fully expanded (10–12 days after leaf appearance), four different salt treatments were applied: (1) control solution, without additional salt in the nutrient solution, (2) NaCl solution, additional 50 mmol NaCl (3) $Cl^-$ dominant, additional 50 mmol $Cl^-$, and (4) $Na^+$ dominant, additional 50 mmol $Na^+$ in the solution. Details on salt compositions between treatments can be found in Table 1. These mixtures of salts were designed to obtain the same osmolarity in all treatments and to ensure that the osmotic effects between treatments were the same. The pH value was adjusted to 6.0–6.2 by 1% sulfuric acid. The nutrient solution was changed once a week. All side shoots and fruits up to the sixth node were removed. During this experiment, the average day/night temperature, relative humidity, and $CO_2$ concentration recorded inside the greenhouse were 25.3 ± 1.4/21.0 ± 0.6 °C, 46.2% ± 10.7%/59.7% ± 8.0%, and 415 ± 26/445 ± 14 ppm, respectively. This experiment used a randomized complete block design with five blocks, arranged along the light gradient in the greenhouse. Each block comprised one plant per treatment. Plant density was 1 plant $m^{-2}$.

**Table 1.** Salt compositions of experiment 1.

| Treatment | Salt Composition | Osmolarity |
|---|---|---|
| Control | Ferty 1 + $Ca(NO_3)_2$ | 0 mOsm $l^{-1}$ |
| 50 mM NaCl | Control + 50 mM NaCl | 100 mOsm $l^{-1}$ |
| 50 mM $Cl^-$ | Control + 9 mM $CaCl_2$ + 20 mM KCl + 6 mM $MgCl_2$ + 7.5 mM $MgSO_4$ | 100 mOsm $l^{-1}$ |
| 50 Mm $Na^+$ | Control + 10 mM $Na_2SO_4$ + 10 mM $NaNO_3$ + 10 mM $Na_2HPO_4$ + 10 mM $MgSO_4$ | 100 mOsm $l^{-1}$ |

*2.2. Experiment 2—Effects of Grafting on Salt Recirculation*

Pumpkin (*Cucurbita moschata* Duchesne, "Becada," Rijk Zwaan, De Lier, The Netherlands) and cucumber seeds ("Aramon," Rijk Zwaan, De Lier, The Netherlands) were sown in rockwool cubes under a 12-h photoperiod (photosynthetic active radiation (PAR) ≈ 300 µmol $m^{-2}$ $s^{-1}$), with 24/20 °C day–night temperature, 70% relative humidity, and 380 ppm $CO_2$ concentration in growth chambers. Four graft combinations were obtained: (1) cucumber without grafting (Cs), (2) pumpkin without grafting (Cm), (3) cucumber grafted onto pumpkin (Cs/Cm), and (4) self-grafted cucumber (Cs/Cs). Pumpkin seeds were sown 3 days ahead of cucumber. The one-cotyledon grafting method [30] was used on day 7 after sowing the cucumber seeds. The cucumber and pumpkin plants for the treatments without grafting were sown two weeks after the scion plants. On 19 May 2014 (21 days after grafting), seedlings were transplanted to the control nutrient solution (see experiment 1) in the greenhouse of the Institute of Horticultural Production Systems, Leibniz Universität Hannover, Germany (52.5°N, 9.7°E). Salinity treatments, 0 and 60 mM additional NaCl in the nutrient solution, were started on day 11 after transplanting to the greenhouse. This experiment used a randomized complete block design with four blocks, arranged along the light gradient in the greenhouse. Each block comprised one plant per treatment. Plant density was 1 plant $m^{-2}$.

### 2.3. Gas Exchange Measurements

Photosynthesis and stomatal conductance of leaves were measured using a portable gas exchange system (Li-6400XT, Li-Cor Inc., Lincoln, NE, USA). During measurements, leaf temperature, $CO_2$ concentration, and light conditions were set at 25 °C, 380 µmol mol$^{-1}$, and 1300 µmol m$^{-2}$ s$^{-1}$, respectively. In experiment 1, measurements were performed on days 1, 4, 8, 11, 15, and 18 after salt application (DAS) between 08:00 and 16:00 h, using the second (1 and 4 DAS), third (8 and 11 DAS), and fourth (15 and 18 DAS) leaves. After gas exchange measurements, the measured leaf parts were harvested and dried for 72 h at 70 °C for chemical analysis. In experiment 2, measurements were performed on 1 and 16 DAS at the first fully expanded leaves under 600 µmol m$^{-2}$ s$^{-1}$ PAR, 25 °C leaf temperature, and 380 µmol mol$^{-1}$ ambient $CO_2$ concentration.

### 2.4. Determination of Plant Growth

On 21 and 23 DAS in experiment 1 and 2, respectively, whole plants were harvested. After harvest, leaf area was measured using a leaf area meter (LI-3100, LI-COR, Inc., Lincoln, NE, USA). Fresh weights and lengths of leaves, petioles, and internodes were recorded. Dry masses of plant materials were weighed after drying at 70 °C for 72 h. In experiment 2, leaves 3, 8, and 15 were sampled separately on 23 DAS for chemical analyses to investigate the effects of grafting on ion distribution in the shoot.

### 2.5. Ion Concentrations in the Xylem Sap

Xylem sap was collected using the root-pressure method [17,31,32]. At the time of harvest, stems were cut ~1 cm below the cotyledons or below the graft (in the grafted plants). The first 100 µL xylem sap exudates from the cut stem were discarded to avoid contamination from phloem sap or cytosolic compounds [31]. The exudates were collected in preweighted Eppendorf tubes containing 20 µL of 1% (*v/v*) 2-mercaptoethanol for 5~15 min, their quantity recorded (>100 µL), and then cool-stored at 5 °C until analyzed for Na$^+$, K$^+$, and Cl$^-$.

### 2.6. Chemical Analysis of Plant Materials

To determine Na$^+$ and K$^+$ concentrations of plant materials, 100 mg dry samples were ground and then ashed at 500 °C in a muffle furnace (Nabetherm L9/11/S27, Lilienthal, Germany). The ashed samples were dissolved in 3 mL nitric acid (1M HNO$_3$) and then 12 mL water. The resulting solution was analyzed with an atom absorption spectrometer (Perkin Elmer 1100B, Waltham, MA, USA). For chloride analysis, 100–200 mg of ground dry samples were dissolved using 40 mL distilled water and heated at 55 °C for 30 min in a water bath. The samples were cooled down for 20 to 30 min and 0.8 mL of 5M HNO$_3$ was added to the solution. For the analyses, the solution was titrated using 0.01M AgNO$_3$ and the chloride concentration was determined using Titro Line$^®$ 6000 (SI Analytics GmbH, Mainz, Germany) when all the chloride was precipitated as silver chloride (AgCl).

### 2.7. Statistical Analyses

All statistical analyses were conducted in R (v.2.12.0; R Foundation for Statistical Computing). Tukey's Honest Significant Differences were calculated to test the differences of parameters between treatments. To estimate the toxic effects of different ions on photosynthesis and stomatal conductance, the following model was applied [11,18]:

$$P_r = (1 + \alpha C_{Na} + \beta C_K + \gamma C_{Cl}) \tag{1}$$

where $P_r$ is photosynthesis rate or stomatal conductance as a fraction of the control, $C_{Na}$, $C_K$, and $C_{Cl}$, are the Na$^+$, K$^+$, and Cl$^-$ concentrations in the leaves (mmol g DW$^{-1}$), respectively, and $\alpha$, $\beta$, and $\gamma$ are the sensitivities of photosynthesis or stomatal conductance to the corresponding ionic components. Models with different combinations of these ionic components were compared by the

Akaike information criterion (AIC). Since the osmotic effect on $P_r$ and the ability of ion exclusion at the leaf level to avoid toxic effects are not considered in the Equation (1) [18,33], an alternative model was also fitted for each ion $x$:

$$P_r = (1 + O_s) \quad \text{for } C_x \leq c \tag{2}$$

$$P_r = (1 + O_s + \kappa C_x) \quad \text{for } C_x \geq c \tag{3}$$

where $O_s$ (%) represents the percentage of osmotic stress on $P_r$, $C_x$ is the concentration of ion $x$, $c$ is the threshold ion concentration below which the effect of the ion $x$ is not significant, and $\kappa$ is the sensitivity of $P_r$ to the ion concentration.

## 3. Results

### 3.1. NaCl and Na⁺ Dominant Solutions Reduced Plant Growth, but Not Cl⁻ Dominant Solution

In the mixed-salt experiment (expt1), plant dry mass (PDM) and total leaf area (TLA) in both cultivars Aramon and Line-759 in NaCl and $Na^+$ dominant solutions were 30–64% lower than in the controls, while those in the $Cl^-$ dominant solution remained the same as the control in both cultivars (Table 2). Line-759 showed a higher PDM reduction in the $Na^+$ dominant solution than Aramon. Interestingly, TLA of Aramon was larger in the $Cl^-$ dominant solution than in the control. Internode lengths, especially of Aramon, were shorter in the salt treatments than in the controls.

**Table 2.** Effects of mixed-salt (expt1) and grafting (expt2) on plant dry mass (PDM), total leaf area (TLA), and internode length (IL, average of ranks 6–15) in experiments 1 and 2.

| | PDM (g Plant⁻¹) | | TLA (m² Plant⁻¹) | | IL (cm) | |
|---|---|---|---|---|---|---|
| **Experiment 1** Treatment | Aramon | Line-759 | Aramon | Line-759 | Aramon | Line-759 |
| Control | 61.7 ± 10.2 | 53.5 ± 10.3 | 1.16 ± 0.03 | 1.05 ± 0.06 | 11.2 ± 0.1 | 13.7 ± 0.7 |
| NaCl | 35.3 ± 2.4 | 37.5 ± 7.9 | 0.89 ± 0.04 | 0.80 ± 0.06 | 9.8 ± 0.2 | 12.0 ± 0.7 |
| Cl⁻ dominant | 56.2 ± 7.3 | 48.9 ± 7.2 | 1.34 ± 0.04 | 1.07 ± 0.07 | 9.8 ± 0.2 | 12.2 ± 0.2 |
| Na⁺ dominant | 24.0 ± 3.7 | 19.0 ± 4.1 | 0.38 ± 0.02 | 0.25 ± 0.03 | 8.6 ± 0.4 | 9.8 ± 0.3 |
| **Significance** | | | | | | |
| Treatment (T) | *** | | *** | | *** | |
| Genotype (G) | ** | | *** | | *** | |
| T × G | ns | | ns | | ns | |
| **Experiment 2** Grafting type | 0 mM | 60 mM | 0 mM | 60 mM | 0 mM | 60 mM |
| Cs | 94.1 ± 15.5 | 28.7 ± 1.6 | 1.11 ± 0.08 | 0.47 ± 0.04 | 8.5 ± 0.1 | 7.2 ± 0.3 |
| Cs/Cs | 100.9 ± 2.1 | 36.7 ± 8.5 | 1.17 ± 0.07 | 0.33 ± 0.10 | 8.5 ± 0.7 | 7.0 ± 1.3 |
| Cm | 99.5 ± 7.2 | 54.1 ± 1.4 | 1.11 ± 0.03 | 0.75 ± 0.01 | 11.9 ± 4.2 | 12.4 ± 1.6 |
| Cs/Cm | 76.7 ± 11.3 | 47.2 ± 8.5 | 0.87 ± 0.04 | 0.56 ± 0.04 | 8.1 ± 0.2 | 7.0 ± 0.2 |
| **Significance** | | | | | | |
| Grafting (G) | ** | | *** | | *** | |
| Salinity (S) | *** | | *** | | * | |
| G × S | *** | | *** | | ns | |

Values are means ± SD ($n = 5$ and $n = 4$ for expt1 and expt2, respectively). Significance between treatments were tested by two-way analysis of variance (ns: nonsignificant; * $p < 0.05$; ** $p < 0.01$; *** $p < 0.001$). Four graft combinations are cucumber without grafting (Cs), pumpkin without grafting (Cm), cucumber grafted onto pumpkin (Cs/Cm), and self-grafted cucumber (Cs/Cs).

In the grafting experiment (expt2), PDM and TLA in the Cs/Cm treatment were about 20% smaller than in other grafting types under control conditions, but the salinity effects on PDM and TLA in the Cs/Cm treatment were smaller than in the other grafting types (Table 2).

### 3.2. Sodium Concentrations in the Xylem Sap Showed Significant Genotypic Difference in Sodium Exclusion

The mixed-salt experiment showed 1.7–2 fold higher $Na^+$ concentrations in the xylem sap of Aramon than in Line-759 under NaCl treatment, indicating genotypic differences in $Na^+$ uptake (Table 3). In contrast, $K^+$ and $Cl^-$ concentrations were similar in both genotypes. $Na^+$ concentrations in the xylem sap in Line-759 under $Na^+$ dominant solutions were tendentially lower than those in Aramon (Table 3). In experiment 2, $Na^+$ concentrations in the xylem sap of pumpkin (Cm and Cs/Cm) were 6.7–9.7% of that of cucumber (Cs and Cs/Cs, Table S1), showing significant difference in $Na^+$ uptake by roots.

**Table 3.** Ion concentration of xylem saps of cucumber genotypes "Aramon" and "Line-759" treated with NaCl, $Cl^-$, and $Na^+$ dominant solution during three weeks. Values are means ± SD ($n$ = 5). Means followed by the same letter are not significantly different at $p$ = 0.05 according to the Tukey Honest Significant Differences test.

| Treatment | $Na^+$ (mM) | | $K^+$ (mM) | | $Cl^-$ (mM) | |
|---|---|---|---|---|---|---|
| | Aramon | Line-759 | Aramon | Line-759 | Aramon | Line-759 |
| Control | 0.3 ± 0.03 [a] | 0.3 ± 0.1 [a] | 16.3 ± 1.5 [a] | 13.2 ± 2.5 [a] | 8.0 ± 1.1 [a] | 8.2 ± 1.7 [a] |
| NaCl | 16.5 ± 2.4 [c] | 9.5 ± 1.9 [b] | 17.0 ± 5.0 [a] | 19.3 ± 2.5 [a] | 37.1 ± 3.3 [b] | 31.9 ± 2.6 [b] |
| $Cl^-$ dominant | 0.6 ± 0.1 [a] | 0.3 ± 0.1 [a] | 39.0 ± 3.9 [b] | 35.4 ± 2.3 [b] | 37.2 ± 5.8 [b] | 36.4 ± 3.8 [b] |
| $Na^+$ dominant | 12.5 ± 1.4 [b] | 7.2 ± 3.0 [b] | 13.6 ± 2.9 [a] | 12.5 ± 3.0 [a] | 9.7 ± 5.5 [a] | 4.1 ± 1.4 [a] |

### 3.3. Mixed-Salt Experiment Creates Large Variations in Patterns of Ion Accumulation and Concentration in Leaves

For both cultivars, leaf $Na^+$ concentrations were similar in NaCl and $Na^+$ dominant solutions (except on 11 DAS in Aramon, Figure 1A,B) and the $Cl^-$ concentrations in leaves were the same in the NaCl and $Cl^-$ dominant solutions (except on 15 DAS in Line-759, Figure 1E,F). In the $Cl^-$ dominant treatment, $Na^+$ concentrations in leaves of both cultivars were 2.2–4.6 fold higher than in the control on 1 and 4 DAS and decreased to the level of the control (Figure 1A,B). In the $Na^+$ dominant solution, $Cl^-$ concentrations in leaves of both cultivars were 34–71% lower than in the control (Figure 1E,F). Although the $Cl^-$ dominant treatment had the highest $K^+$ concentrations in the nutrient solution (Table 1) and in the xylem sap (Table 3), $K^+$ concentrations in leaves of the $Cl^-$ dominant treatment were 17–26% lower than in the control, and similar to or slightly higher than in the NaCl and $Na^+$ dominant solutions (Figure 1C,D). In both cultivars, $Na^+$ accumulation rates in the NaCl and $Na^+$ dominant solutions and $Cl^-$ accumulation rates in the NaCl and $Cl^-$ dominant solutions were not different. Under salt treatments, Aramon accumulated more $Na^+$ and $Cl^-$ than Line-759 (Figure 1A,B,E,F), while $K^+$ concentrations of the two cultivars were similar (Figure 1C,D).

The differences in $Na^+$ and $Cl^-$ accumulation rates between treatments and cultivars resulted in a wide range of $K^+/Na^+$ and $Na^+/Cl^-$ ratios in the leaves (Figure 2). The relationships between $Na^+$ and $K^+$ were not different between NaCl and $Na^+$ dominant solutions for both Aramon (Figure 2A) and Line-759 (Figure 2B). The $K^+$ concentrations in the leaves of the two cultivars did not differ greatly, but Aramon accumulated up to twice as much $Na^+$ in the leaves than Line-759 (Figure 1A,B and Figure 2A,B). This large variation allowed the use of statistical models to quantify the potential effects of a specific ion on photosynthesis and stomatal conductance (Figure 3).

### 3.4. Photosynthesis and Stomatal Conductance Were Sensitive to $Na^+$, Not to $K^+$ and $Cl^-$

Using Equation (1) to fit the relationships between photosynthesis and ion concentrations showed that, for the genotype Aramon, the model considering only $C_{Na}$ had the best fit ($R^2$ = 0.45, $p$ < 0.0001) and the parameters $\beta$ and $\gamma$, the coefficients of $C_K$ and $C_{Cl}$, were not significant ($p$ = 0.37 and 0.50, respectively). Therefore, the model considering all ionic components was statistically not better than

($p = 0.65$) the model considering only $C_{Na}$. The value of $\alpha$ in the model considering only $C_{Na}$ was $-0.77 \pm 0.11$ ($p < 0.0001$).

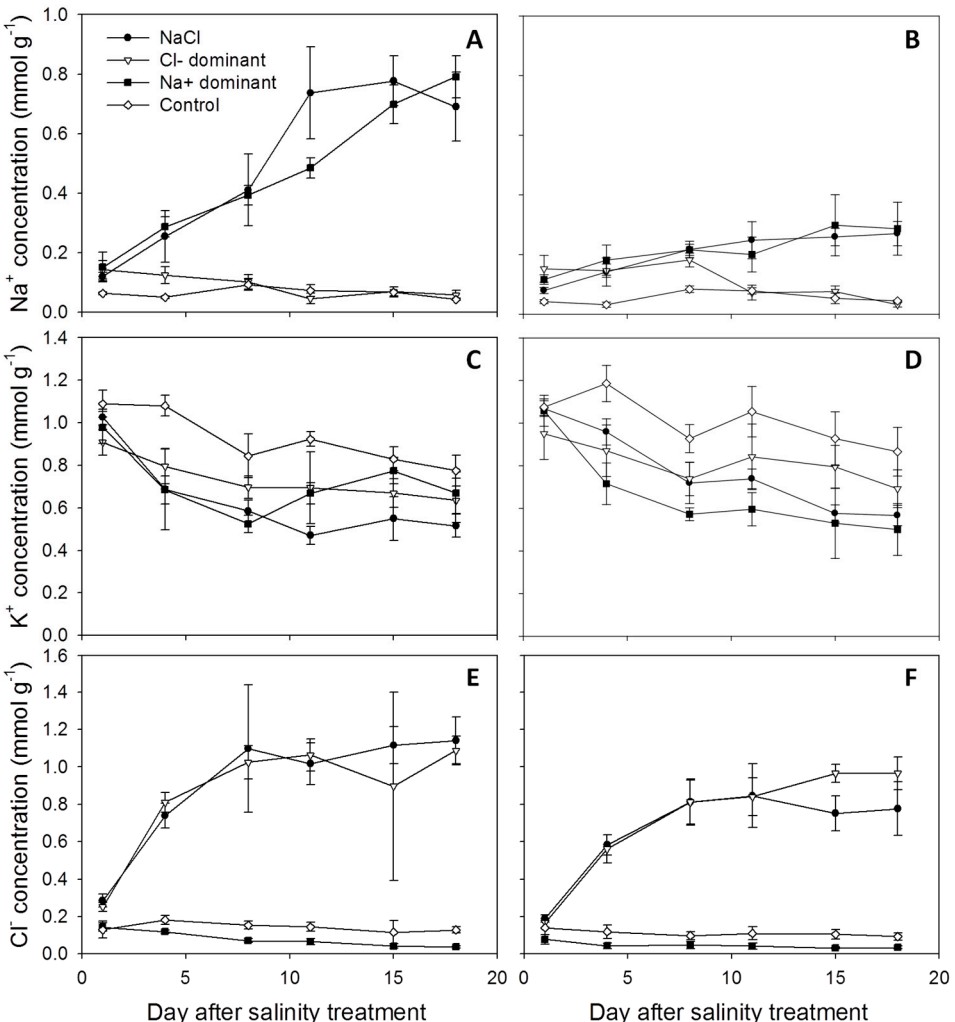

**Figure 1.** Concentrations of Na$^+$ (**A,B**), K$^+$ (**C,D**) and Cl$^-$ (**E,F**) in leaves of cucumber genotypes Aramon (A,C,E) and Line-759 (B,D,F) grown under different salt combinations. Samples were obtained from the second (1 and 4 DAS), third (8 and 11 DAS), and fourth (15 and 18 DAS) leaves. Values are means ± SD ($n = 5$).

Similar to Aramon, $C_{Na}$ explained the largest part of the variation in photosynthesis rate and stomatal conductance of Line-759 (Figure 3D). For photosynthesis rate, the model considering $C_{Na}$ and $C_K$ had the best fit ($R^2 = 0.44$, $p < 0.0001$) but the parameter $\beta$ describing the effect of $C_K$ was not significant ($p = 0.17$). In Line-759, $\alpha$ in the model considering only $C_{Na}$ was $-1.62 \pm 0.26$ ($p < 0.0001$), showing a more sensitive response of photosynthesis to Na$^+$ toxicity. For both cultivars, the influence of $C_{Cl}$ on photosynthesis and stomatal conductance was not significant, although a 20–40% decrease in photosynthesis rate was observed in plants grown in the Cl$^-$ dominant solution (Figure 3C,F). This decrease was partly due to the fact that osmotic effects on $P_r$ were not considered in Equation (1). By fitting photosynthesis rate of Aramon and Line-759 to Equation (2b) with $C_{Na}$, osmotic effects ($O_s$) on photosynthesis were $-0.134 \pm 0.071$ ($p < 0.0001$) and $-0.107 \pm 0.069$ ($p = 0.14$), threshold Na$^+$ concentrations $c$ were $0.260 \pm 0.143$ ($p = 0.09$) and $0.170 \pm 0.032$ ($p < 0.0001$), and sensitivities $\kappa$ were $-0.942 \pm 0.322$ ($p = 0.01$) and $-4.298 \pm 1.498$ ($p = 0.01$), respectively. Equation (3) fitted with

$C_{Na}$ predicted photosynthesis rate slightly better than Equation (1) ($R^2 = 0.53$ and 0.54, respectively). Furthermore, when fitting Equation (3) with $C_{Cl}$ and $C_K$, the model did not converge.

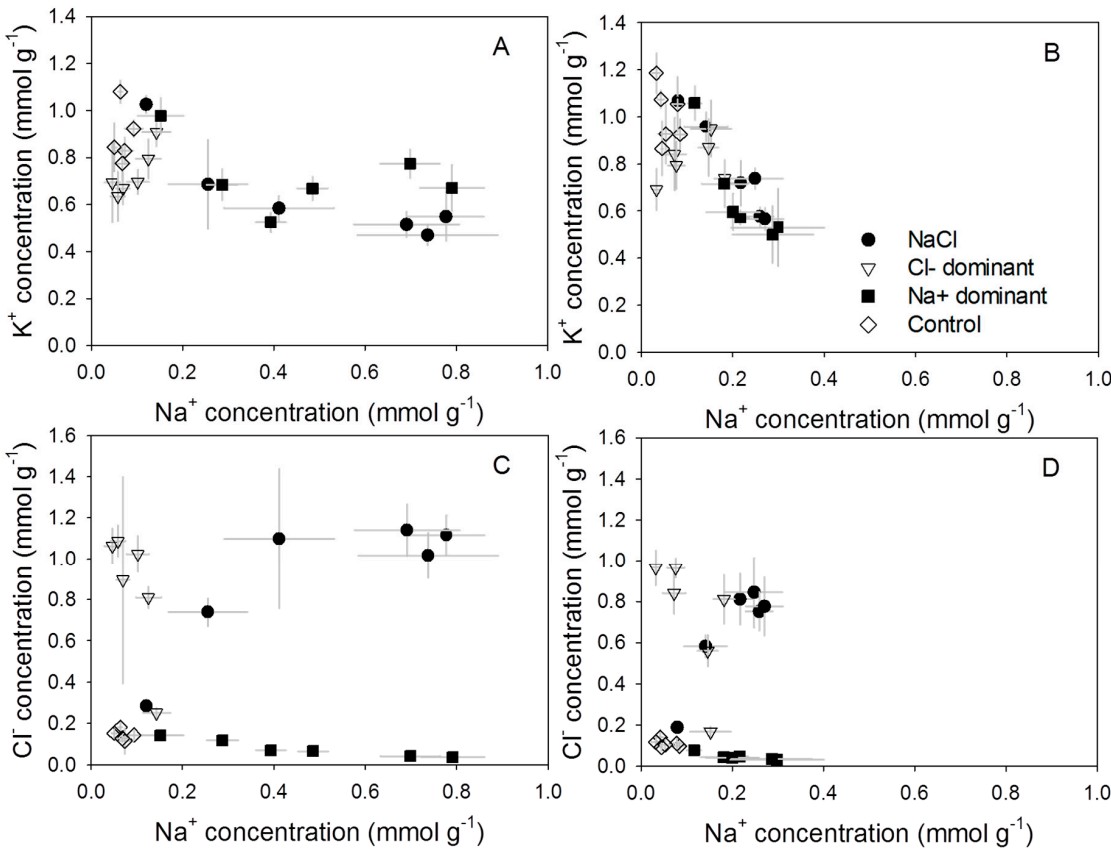

**Figure 2.** Relationships between ion concentrations in the leaves of cucumber genotypes Aramon (**A**,**C**) and Line-759 (**B**,**D**) in experiment 1. Values are means ± SD (*n* = 5).

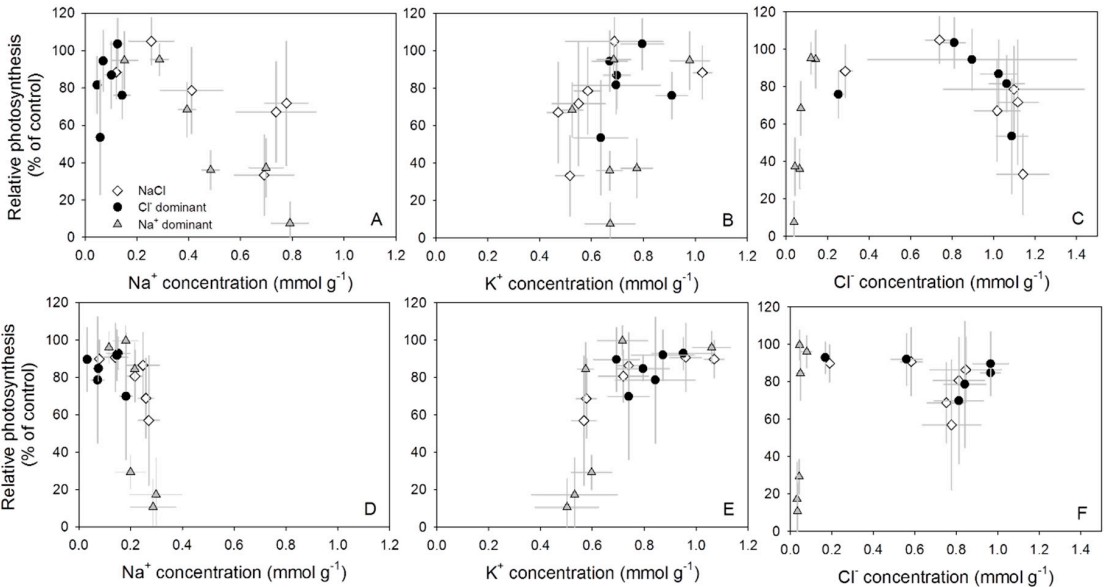

**Figure 3.** Relationships between ion contents in the leaves (Na$^+$ in **A**,**D**; K$^+$ in **B**,**E**; Cl$^-$ in **C**,**F**) and relative photosynthesis (% of control plants) of cucumber genotypes Aramon (A,B,C) and Line-759 (D,E,F) in experiment 1. Values are means ± SD (*n* = 5).

For both cultivars, the models fitted to stomatal conductance had similar results since the dependency of photosynthesis to stomatal conductance was not different between treatments (Figure S1).

### 3.5. Pumpkin Improved Photosynthesis and Regulated Ion Distributions in Plants through Recirculation

$Na^+$ and $Cl^-$ concentrations in leaves were not different between grafting types and between young (leaf 15) and old (leaf 3) leaves under nonstressed conditions (Table 4). In contrast, Cm and Cs/Cm accumulated less than 10% of $Na^+$ in comparison to Cs and Cs/Cs under salinity and the $Na^+$ concentrations in young leaves tended to be lower than in older leaves. Under nonstressed conditions, $K^+$ concentration in leaf 15 was less than (in Cs treatment) or similar to (in Cs/Cs, Cm and Cs/Cm treatments) old leaves. $K^+$ concentrations in leaves were lower under salinity than in the controls. Interestingly, $K^+$ concentrations in the young leaves of Cm and Cs/Cm treatments were about two times higher than in old leaves under salinity (Table 4), suggesting that pumpkin rootstock improved $K^+$ recirculation to the young leaves. In contrast, more $Cl^-$ (1.9–2.8 fold) was accumulated in the old leaves of Cm and Cs/Cm treatments than in young leaves, while ratios of $Cl^-$ concentrations between old and young leaves of Cs and Cs/Cs were similar. This indicated an active $Cl^-$ recirculation towards old leaves through pumpkin roots. In all treatments, ion concentrations of leaf 8 were the same as in the rest of the leaves (Table S2). Grafting did not affect photosynthetic capacity of cucumber leaves since on 1 DAS, leaf photosynthesis rate was unaffected by salinity and by grafting (Figure 4A). On 16 DAS, photosynthesis rates of Cm and Cs/Cm treatments under salinity were as high as in control plants, but a decrease in photosynthesis of around 65% due to salinity was measured in Cs and Cs/Cs treatments (Figure 4B).

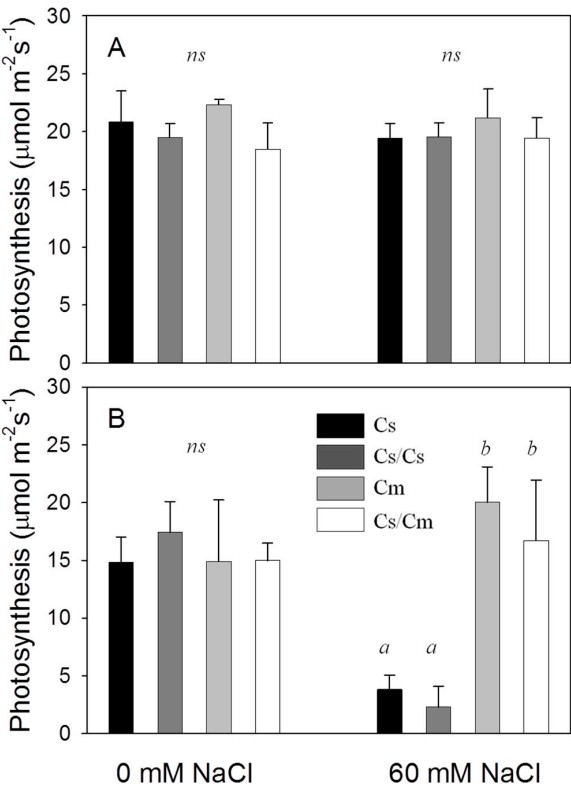

**Figure 4.** Effects of grafting and salinity on leaf photosynthesis rate measured on the youngest fully expanded leaves in experiment 2 on (**A**) 1 DAS and (**B**) 16 DAS (*n* = 4). Four graft combinations are cucumber without grafting (Cs), pumpkin without grafting (Cm), cucumber grafted onto pumpkin (Cs/Cm), and self-grafted cucumber (Cs/Cs). Means followed by the same letter are not significantly different at *p* = 0.05 according to the Tukey Honest Significant Differences test.

**Table 4.** Influence of grafting on the ion concentrations in different leaves on day 23 after salinity treatment.

| Ion | Salinity Level | Grafting Type | Leaf 3 (mmol g$^{-1}$) | Leaf 8 (mmol g$^{-1}$) | Leaf 15 (mmol g$^{-1}$) |
|---|---|---|---|---|---|
| Na$^+$ | 0 mM NaCl | Cs | 0.06 ± 0.01 $^a$ | 0.04 ± 0.01 $^a$ | 0.09 ± 0.03 $^a$ |
| | | Cs/Cs | 0.05 ± 0.02 $^a$ | 0.06 ± 0.01 $^a$ | 0.07 ± 0.00 $^a$ |
| | | Cm | 0.05 ± 0.02 $^a$ | 0.05 ± 0.02 $^a$ | 0.06 ± 0.01 $^a$ |
| | | Cs/Cm | 0.09 ± 0.03 $^a$ | 0.03 ± 0.00 $^a$ | 0.07 ± 0.02 $^a$ |
| | 60 mM NaCl | Cs | 1.25 ± 0.19 $^{bc}$ | 1.47 ± 0.12 $^c$ | 1.36 ± 0.09 $^{bc}$ |
| | | Cs/Cs | 1.07 ± 0.12 $^b$ | 1.19 ± 0.10 $^{bc}$ | 1.27 ± 0.62 $^{bc}$ |
| | | Cm | 0.08 ± 0.04 $^a$ | 0.06 ± 0.00 $^a$ | 0.03 ± 0.01 $^a$ |
| | | Cs/Cm | 0.07 ± 0.01 $^a$ | 0.06 ± 0.02 $^a$ | 0.04 ± 0.01 $^a$ |
| K$^+$ | 0 mM NaCl | Cs | 0.74 ± 0.07 $^{ab}$ | 0.61 ± 0.06 $^{bcdef}$ | 0.38 ± 0.05 $^{ghi}$ |
| | | Cs/Cs | 0.65 ± 0.06 $^{abcd}$ | 0.63 ± 0.03 $^{bcde}$ | 0.49 ± 0.06 $^{cdefghi}$ |
| | | Cm | 0.62 ± 0.08 $^{bcde}$ | 0.66 ± 0.09 $^{abc}$ | 0.63 ± 0.08 $^{bcde}$ |
| | | Cs/Cm | 0.56 ± 0.06 $^{bcdefg}$ | 0.56 ± 0.04 $^{bcdefg}$ | 0.57 ± 0.05 $^{bcdefg}$ |
| | 60 mM NaCl | Cs | 0.45 ± 0.09 $^{efghi}$ | 0.45 ± 0.09 $^{defghi}$ | 0.46 ± 0.12 $^{defghi}$ |
| | | Cs/Cs | 0.46 ± 0.06 $^{defghi}$ | 0.31 ± 0.00 $^i$ | 0.29 ± 0.07 $^i$ |
| | | Cm | 0.35 ± 0.02 $^{hi}$ | 0.52 ± 0.06 $^{cdefgh}$ | 0.83 ± 0.03 $^a$ |
| | | Cs/Cm | 0.36 ± 0.01 $^{hi}$ | 0.41 ± 0.02 $^{fghi}$ | 0.68 ± 0.06 $^{abc}$ |
| Cl$^-$ | 0 mM NaCl | Cs | 0.25 ± 0.03 $^g$ | 0.25 ± 0.06 $^g$ | 0.19 ± 0.04 $^g$ |
| | | Cs/Cs | 0.25 ± 0.04 $^g$ | 0.29 ± 0.03 $^g$ | 0.27 ± 0.06 $^g$ |
| | | Cm | 0.67 ± 0.03 $^f$ | 0.41 ± 0.03 $^{fg}$ | 0.24 ± 0.02 $^g$ |
| | | Cs/Cm | 0.46 ± 0.08 $^{fg}$ | 0.33 ± 0.11 $^g$ | 0.24 ± 0.06 $^g$ |
| | 60 mM NaCl | Cs | 1.51 ± 0.07 $^{bc}$ | 1.69 ± 0.15 $^{ab}$ | 1.37 ± 0.17 $^{cd}$ |
| | | Cs/Cs | 1.12 ± 0.22 $^{de}$ | 1.37 ± 0.04 $^{cd}$ | 1.45 ± 0.12 $^{bc}$ |
| | | Cm | 1.94 ± 0.06 $^a$ | 1.42 ± 0.1 $^{bc}$ | 0.99 ± 0.04 $^e$ |
| | | Cs/Cm | 1.82 ± 0.09 $^a$ | 1.41 ± 0.07 $^{bcd}$ | 0.65 ± 0.16 $^f$ |

Leaf 3 represents the older leaf and leaf 15 represents the youngest fully expanded leaf. Values are means ± SD (*n* = 3). Four graft combinations are cucumber without grafting (Cs), pumpkin without grafting (Cm), cucumber grafted onto pumpkin (Cs/Cm) and self-grafted cucumber (Cs/Cs). Means followed by the same letter are not significantly different at *p* = 0.05, according to the Tukey Honest Significant Differences test.

## 4. Discussion

### 4.1. Photosynthetic Capacity of Cucumber Is More Sensitive to Na$^+$ Than Cl$^-$

Using two cucumber cultivars with different abilities to exclude Na$^+$ and Cl$^-$ and using different salt mixtures (Table 1), we generated various patterns of ion concentrations and ratios in the leaves (Figure 2) for testing the toxic effects of different ions on leaf photosynthesis and stomatal conductance (Figure 3). To evaluate the ionic effects on ecophysiological traits (e.g., photosynthesis and stomatal conductance) by a mixed-salt experiment, two prerequisites have to be fulfilled. First, the osmotic stress between mixed-salt treatments has to be equal (Table 1) since the strength of the osmotic stress affects stomatal closure and then photosynthesis at the leaf level, and total leaf area at the whole plant level [1]. Furthermore, osmotic stress may affect ion accumulation in leaves by reducing transpiration [19]. This explains that Cl$^-$ concentrations in both cultivars were higher under control conditions than in Na$^+$ dominant solutions (Figure 1E,F). Second, the Na$^+$ concentrations in the NaCl and Na$^+$ dominant treatments and the Cl$^-$ concentration in the NaCl and Cl$^-$ dominant treatments have to be the same since the ion concentrations in the growth medium strongly affect ion uptake and consequently ion accumulation rates in the leaves [23–25]. Our results show that Na$^+$ accumulation rates of leaves in the NaCl and Na$^+$ dominant solutions (Figure 1A,B) and Cl$^-$ accumulation rates in the NaCl and Cl$^-$ dominant solutions (Figure 1E,F) were the same in both cultivars. This assured that the leaves between treatments and control were comparable in age and ion concentrations, the prerequisite for attributing response curves of photosynthesis and stomatal conductance to ion concentrations [18]. To construct the response curves, photosynthesis and stomatal conductance were expressed in relation

to control plants in order to normalize for the effects of leaf age [18]. Cucumber has a high leaf turnover rate (the biochemical capacity for photosynthesis drops by 50% after 15 days past full expansion of the leaf [11,34,35]) and the accumulation of salt to a toxic level takes 15–20 days [11]. Without this normalization the ionic effects would be overestimated.

Our results from leaf level and whole plant level suggest that, in both cultivars, photosynthesis and stomatal conductance were sensitive to $Na^+$ concentration but insensitive to $Cl^-$ in the leaf (Figure 3). This explains the strong decreases in plant dry mass that occurred in NaCl and $Na^+$ dominant solutions, but not under $Cl^-$ dominant solution (Table 2). Interestingly, total leaf area in the $Cl^-$ dominant solution was not affected while total plant dry mass was about 10% lower than that of the control (Table 2). This indicates that osmotic effects of the $Cl^-$ dominant solution on leaf expansion and light interception were not prominent and the differences in plant dry mass between the control and $Cl^-$ dominant treatment mainly resulted from the osmotic effects on leaf photosynthesis, which reduced photosynthesis rate by 13% and 11% in Aramon and Line-759, respectively, according to the value of $O_s$ in Equations (2) and (3). Since $Cl^-$ can be used by the plant as an osmoregulator to promote leaf expansion, the insignificant difference in total leaf area between the control and $Cl^-$ dominant solution was probably a counterbalance between the negative effects of osmotic stress and the positive effects of $Cl^-$ on leaf cell expansion [36,37]. It is also interesting that cucumber seems to have mechanisms avoiding overaccumulation of $Cl^-$. In both cultivars, $Cl^-$ accumulation rates between 8 and 18 DAS were close to zero (Figure 1E,F) while $Na^+$ was still accumulated at rates similar to those between 1–8 DAS (Figure 1A,B). Similar phenomena were not observed in experiments with cucumber, where $Na^+$ concentration in leaves was measured only between 1–10 DAS [38], and with maize [39]. In contrast, diverse responses were reported in bean (*Phaseolus vulgaris* L.), a $Cl^-$ sensitive species, and in brown beetle grass (*Diplachne fusca*), a halophyte with salt glands, that $Cl^-$ in both species can be further accumulated in leaves while the $Na^+$ accumulation ceases [26,40].

Results from the grafting experiments provide further evidence that cucumber leaves may tolerate a high amount of $Cl^-$. The young leaves of Cs/Cm grown under salinity stress maintained their photosynthesis rate and $Na^+$ concentration at the same level as the leaves of nonstressed plants (Figure 4), while considerable amounts of $Cl^-$ were accumulated in these leaves (e.g., Table 4). Under salinity, old leaves of Cs/Cm plants accumulated $Cl^-$ up to 1.82 mmol $g^{-1}$ (Table 4) without having visible symptom of ion toxicity (Figure S2). In contrast, the old leaves of Cs and Cs/Cm treatments accumulated a significantly lower amount of $Cl^-$ than Cs/Cm and were already desiccated (Figure S2). Interestingly, $Cl^-$ concentration of Cs/Cm plants rose by about 50% above the maximum value obtained from experiment 1 (for NaCl and $Cl^-$ dominant solutions, Figure 1E). This could imply that Cs/Cm plants have a better tissue tolerance to $Cl^-$ than nongrafted plants, probably induced by the hormones, proteins, or messenger RNAs produced by the rootstock [30]. This remains an interesting hypothesis for future study.

### 4.2. Ion Exclusion and Tissue Tolerance Vary Independently in Cucumber

Ion exclusion and tissue tolerance are important mechanisms of salinity tolerance [9,41–44]. Line-759 can exclude $Na^+$ more effectively than cultivar Aramon, since the $Na^+$ concentrations in its xylem sap were significantly lower than in Aramon (Table 3). In Line-759, a higher sensitivity of photosynthesis and stomatal conductance to $Na^+$ concentration (a more negative $\alpha$ of Equation (1) and $\kappa$ of Equation (3)) indicates its weaker tissue tolerance to $Na^+$ compared to Aramon (Figure 3A,D). A higher value of $\alpha$ and $\kappa$ can be achieved by the plants' ability to compartmentalize the toxic ions in the vacuole [28] or a better adjustment of leaf water and osmotic potentials during ion accumulation [28,45,46]. Since the water and osmotic potentials of the leaves were not different between genotypes and salt treatments during the experiment (Figure S3), it is plausible that this difference in tissue tolerance is due to ion compartmentalization at the cellular and subcellular levels. These results suggest that the mechanisms of ion exclusion of root and tissue tolerance of leaves also vary independently in cucumber, as in bread wheat [9].

Under salinity, the old leaves of Cm and Cs/Cm accumulated surprisingly low amounts of $Na^+$ (in comparison with the values reported in the literature, e.g., [14–16]). This indicates that the pumpkin cultivar Becada used in this study has a distinguished ability to prevent the influx of $Na^+$ into leaves by reducing ion uptake by the roots [6,14–17,23,47,48] or preferential accumulation in the stem or petiole [49,50]. $Na^+$ concentrations in the xylem sap suggest that cucumber roots took up 10–20 times more $Na^+$ than the pumpkin cultivar Becada under salinity (Table S1). Consequently, $Na^+$ concentrations in cucumber leaves were also 10–20 times higher (Table 4 and Table S2). Genotypic variation in ion exclusion was found in cucumber (Chen et al., unpublished results), but not for pumpkin. Future work should test the genotypic variation of ion exclusion in pumpkin if grafting onto pumpkin is the most effective horticultural practice of improving salinity tolerance of cucumber. Furthermore, it would be interesting to graft Line-759 onto pumpkin rootstock, or to graft Aramon onto Line-759, to see if tissue tolerance of $Na^+$ is rootstock-dependent.

### 4.3. Pumpkin Rootstock Enhances $K^+$ and $Cl^-$ Recirculation under Salinity

Ion accumulation and ion transport are complex processes associated with a variety of physiological phenomena, such as transpiration, the ability to avoid/enhance ion entering the shoots [42,45–47,51], preferential accumulation of $Na^+$ in supporting tissues or in the vacuole [44,49,52], and recirculation of ions from the shoot to the roots via the phloem [7,8,25]. Results from the grafting experiment do not support the hypothesis that grafting onto pumpkin rootstock enhances $Na^+$ recirculation in the cucumber scion, but surprisingly, grafting onto pumpkin facilitates $K^+$ transport towards the young leaves and enhances $Cl^-$ recirculation from the young to the old leaves (Table 4). This might be explained by a recent study showing that the abilities to remobilize mineral nutrients vary strongly between plant species and depend on the nutrient availability of the growth medium [53]. Under salinity, photosynthesis rate of young leaves is more restricted by the ionic effects on stomatal regulation [11]. Enhancing $K^+$ transport toward the young leaves through grafting would be an acquired mechanism of maintaining $K^+/Na^+$ ratio and stomatal regulation of young leaves.

To have an overall picture how a specific ion is accumulated and transported on the plant level requires the understanding of a complex system [45,51]. A series of modeling studies following the approach proposed by Wolf and Jeschke (1987) investigating ion distribution under salinity can be found in the literature [24,25,54–56]. To our knowledge, this model is surprisingly the only one in the literature that describes ion transport on the whole plant level. This empirical model allows the quantification of the ion fluxes between plant organs and the long-term increments of ion contents in these organs. To combine this model approach with grafting and mixed-salt experiments in the future would provide knowledge about the regulation of ion transport by the rootstock.

## 5. Conclusions

Using mixed-salt experimentation and grafting cucumber onto pumpkin rootstocks, we obtained diverse patterns of $Na^+$ and $Cl^-$ concentrations and ratios in leaves to derive the response curves of ecophysiological traits, e.g., photosynthesis and stomatal conductance, to ion concentrations. We did not find evidence supporting that, in cucumber, $Cl^-$ would be accumulated to a toxic level in leaves. Apparently, there are mechanisms avoiding overaccumulation of $Cl^-$. Mechanisms of $Na^+$ exclusion and tissue tolerance vary independently in the selected cucumber cultivars. Furthermore, our data do not support the hypothesis that grafting onto pumpkin rootstocks facilitates $Na^+$ recirculation in the cucumber scion, but they suggest that grafting enhances $K^+$ transport toward the young leaves and $Cl^-$ transport toward the old leaves.

**Supplementary Materials:** The following are available online at http://www.mdpi.com/2073-4395/10/5/677/s1, Figure S1: Relationships between relative photosynthesis rate and stomatal conductance (% of control plants) in cucumber cultivar Aramon (**A**) and Line-759 (**B**) in experiment 1. Figure S2. Old cucumber leaves on the day of plant harvest (23 DAS) in experiment 2. (**A**) ungrafted cucumber under salinity stress and (**B**) cucumber grafted onto pumpkin under salinity stress. Figure S3. Water potential (**A**,**B**) and osmotic potential (**C**,**D**) of

leaves in genotype Aramon (**A**,**C**) and Line-759 in experiment 1. Table S1: Salt concentrations in the xylem sap in experiment 2, Table S2: Salt concentrations in the rest of the leaves in experiment 2.

**Author Contributions:** Conceptualization, T.-W.C. and H.S.; data collection and analyses, I.M.G.P., A.M.B., and T.-W.C.; writing—original draft preparation, I.M.G.P. and T.-W.C.; writing—review and editing, T.-W.C. and H.S. All authors have read and agreed to the published version of the manuscript.

**Funding:** The publication of this article was funded by the Open Access Fund of the Leibniz Universität Hannover.

**Acknowledgments:** The authors thank USDA–North Central Regional Plant Introduction Station (Ames, IA, USA), who provided the seeds of the cucumber cultivar Line-759. I.M.G.P. was supported by the German Academic Exchange Service (DAAD) through its Development-Related Postgraduate Courses. The authors also thank Ilona Napp for her help throughout the experiment and Marie-Luise Lehmann for chemical analyses.

**Conflicts of Interest:** The authors declare no conflict of interest.

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
