# Peer review of "Determining Ion Toxicity in Cucumber under Salinity Stress"

_agronomy, doi:10.3390/agronomy10050677_

Round 1

Reviewer 1 Report

The study with cucumber and salinity is well designed, planned and executed. However, there are few recommendations before to improve the manuscript. 

Authors can present their results with better figures, especially colored ones. For example, Fig 2 and Fig 3 are hard to understand.

While presenting results, instead of just increase and decrease, it would be good to state how many fold increase or decrease.

Also, it would be good if they could conclude and summarize the paper with a mechanistic model with salinity and difference in these two cultivars and its relation to phothosynthesis and ion distribution.  

Author Response

We have many thanks to these valuable comments. We are especially sorry that Fig. 2 and Fig. 3 were not clear enough. In the revised ms, they were remade, the symbols were enlarged and the error bars became grey to increase the readability of the figures.

We do agree with the suggestion of the reviewer, so the results were rephrased with concrete number (e.g. l192, l196, l207, l209, l212, l259).

Although our data provide some insights into the mechanism of ion transport, the regulation of these processes is highly complex. We feel uncomfortable and too daring to propose a “mechanistic model” from our current data, since the experiments were only designed for determination of ion toxicity.

Reviewer 2 Report

Line 109 – please also include SD along with average value.

Line 110 – RCBD or CRD, what was the blocking factor in the experiment if the design was RCBD.

Experiment 2 – Please clarify how experimental design was RCBD.

Author Response

We have many thanks to these valuable comments. By checking the SD in l109, we found that the day/night CO2 concentrations were reversed. We correct this mistake. Thank you. In the revised ms, we also explicate RCBD in the experiment 1 and 2 (l110, l129).